# National Library of Latvia Subject Headings as Linked Open Data

Mārīte Apenīte [0000-0003-0036-5029] and Uldis Bojārs [0000-0001-7444-565X]

National Library of Latvia, Mūkusalas iela 3, Riga, LV-1423, Latvia
`marite.apenite@lnb.lv, uldis.bojars@lnb.lv`

**Abstract.** This paper presents the linked data representation of the National Library of Latvia (NLL) authority data consisting of topical, geographic name and form/genre thesauri. It is an important step from a data silo (MARC 21 data) to linked open datasets, represented using SKOS, that can be reused both within and outside the library domain. The datasets are converted to SKOS using mc2skos and are published as linked data using the Skosmos application for publishing SKOS datasets. The paper describes the datasets, including information about their external links. The NLL topical terms dataset, in particular, contains many external links – 85% of its concepts have links to the Library of Congress subject headings dataset that is available as linked open data. The published datasets can be applied to interconnecting Latvia's cultural heritage information by describing museum and archive items with URIs from these datasets.

**Keywords:** authority control, linked data, SKOS, National Library of Latvia Subject Headings (NLLSH)

## 1 Introduction

Library authority files are controlled vocabularies that provide standard names and identifiers for different types of headings[1] such as personal names, corporate bodies, geographic names, topical headings, and form or genre terms that library catalogs may need to refer to in a unified way. Authority control plays an important role in library information systems. Authority data ensures: 1) common use of terms, including synonyms, quasi-synonyms, and homonyms; 2) compatibility of metadata with other international controlled vocabularies.

The National Library of Latvia Subject Headings (NLLSH) have been developed and maintained since 2000 as a fundamental and universal vocabulary that forms the basis of the National Thesaurus of Latvia. NLLSH contains topical headings, geographic names, and form/genre terms. Most of topical headings are adopted from the

---

[1] Heading is the name of a person, corporate body, or geographic location; the title proper of a work; or an authorized content descriptor (subject heading), placed at the head of a catalog entry or listed in an index. In library cataloging, genre/form terms are also used. http://library.wcsu.edu/people/reitz/ODLIS/odlis%20new.html

Library of Congress Subject Headings (LCSH)[2] [4] according to Latvian language and terminology. The form/genre terms are adopted from the Faceted Application of Subject Terminology (FAST)[3] and Library of Congress Genre/Form terms (LCGFT)[4]. NLLSH has a thesaurus-like structure – it contains thesaurus relationships between terms: Broader Term (BT), Narrower Term (NT), Related Term (RT), and Used for (UF).

NLLSH authority records ensure a qualitative use of terms in the NLL electronic catalog and special collections, including digital collections. NLLSH authority data are also used by scientific, special, and public libraries in Latvia. NLLSH authority records are created according to international standards and are usable both nationally and internationally. They provide unified subject access and retrieval of various information resources in Latvian, English and other languages.

The development of information technologies, the diversity and digitization of library information resources, and the introduction of linked data are also introducing changes to authority data at NLL. This paper presents the linked data representation of the National Library of Latvia authority data consisting of topical heading, geographic name, and form/genre thesauri.

## 2  NLLSH Linked Data

In 2019, NLL began to enrich NLLSH authority records with external links to other linked data sources. The NLL topical terms dataset, in particular, contains many external links – 85% of its concepts have links to the LCSH dataset that is available as linked open data. The geographic name dataset contains external links to the Place Names Database of Latvia[5] (11% of concepts), Wikidata (5.3%) and Virtual International Authority File dataset (VIAF, 3.5% of concepts) [2]. The form/genre dataset contains external links to the FAST Linked Data dataset (21% of concepts), the Czech National Authority Database (11.7%) and the Library of Congress Genre/Form dataset (8%).

Examples of NLLSH authority records that are too specific and can not be found in LCSH or Wikidata are "Teņa diena" (Tenis' day, related to Latvian traditions and folklore) and "Composite amber threads" (specific to research publications).

The path from MARC 21 authority records to NLL authority linked data consists of three steps:
1) Enriching MARC 21 records;
2) Converting MARC records to SKOS concepts;
3) Publishing SKOS datasets as linked open data.

[2] https://id.loc.gov/authorities/subjects.html
[3] http://fast.oclc.org/searchfast/
[4] https://id.loc.gov/authorities/genreForms.html
[5] http://vietvardi.lgia.gov.lv/

## 2.1    Enriching MARC 21 Data

The original dataset (in MARC 21 format) contains thesaurus relationships between its terms but these links use authority records' primary labels as keys. As a result, these links would be lost in the conversion process described in the next step, which expects records to be linked using authority record numbers. To preserve these links, we enriched MARC 21 records by adding unique record identifiers – record system numbers (MARC field 001) – whenever the record contained links to other records in the dataset. The original authority records created before 2020 do not contain information about the language of concept labels (preferred and alternative labels) which is an obstacle to the correct display of these labels in Skosmos. In order to resolve this shortcoming, we assigned language code "lv" (Latvian) to all preferred labels. This fixes the issue but may not always correctly match the labels' language. It remains future work to automatically determine label language.

## 2.2    Converting MARC 21 Data to SKOS

The datasets are converted to SKOS using the mc2skos application[6] that takes MARC/XML data as input and produces SKOS concepts in Turtle syntax. In order to interlink the resulting datasets mc2skos calls Skosify to infer skos:narrower and skos:related links[7]. Concepts are given URIs based on scheme parameters in mc2skos configuration using the following URI prefixes:

- Topical terms: http://dati.lnb.lv/onto/subject/
- Geographic names: http://dati.lnb.lv/onto/geo/
- Form/genre terms: http://dati.lnb.lv/onto/genre/

The resulting datasets contain 44'407 concepts for topical terms, 9'671 concepts for geographic names and 1'662 concepts for forms/genres.

Links from NLLSH to other datasets are represented using skos:closeMatch following the example of LCSH which represents most of its external links using this property. We decided not to use skos:exactMatch because the concepts in different datasets may change in time and as a result might not be exact matches any more.

## 2.3    Publishing SKOS datasets

The datasets[8] are published using Skosmos – a web-based tool for browsing and publishing SKOS data [5]. According to Skosmos installation instructions[9] RDF data is loaded into a Fuseki RDF store and the resulting Skosmos concept pages are published

---

[6]    https://github.com/scriptotek/mc2skos/
[7]    https://skosify.readthedocs.io/en/latest/
[8]    http://dati.lnb.lv/onto/lv/
[9]    https://github.com/NatLibFi/Skosmos/wiki/InstallTutorial

on the web. Skosmos offers both a human-friendly view as an HTML page, machine-readable RDF data via its API and access to linked data using HTTP content negotia-tion[10]. We created a custom Skosmos plugin that enriches HTML pages with links to the original ALEPH library information system authority records.

Skosmos also shows information about concepts' external links and automatically retrieves labels of the linked concepts published as linked data (e.g. Library of Congress Subject Headings). Figure 1 displays an example of a Skosmos concept page with a link to LCSH linked data.

| PREFERRED TERM | **Ontoloģijas (informācijas izguve) (lv)** |
| --- | --- |
| BROADER CONCEPT | Datu struktūras (datorzinātne) (lv) |
| ENTRY TERMS | *Ontologies (Information retrieval)* |
| NOTE | Source: Ilustrētā svešvārdu vārdnīca, 2005: |
| IDENTIFIER | LNC10-000123185 => ALEPH authority record |
| IN OTHER LANGUAGES | *Ontologies (Information retrieval)* **Ontoloģijas (informācijas izguve)** Latvian |
| URI | http://dati.lnb.lv/onto/subject/LNC10-000123185 |
| Download this concept: | RDF/XML TURTLE JSON-LD    Created 10/29/08, last modified 9/30/15 |
| CLOSELY MATCHING CONCEPTS | Ontologies (Information retrieval)    Library of Congress Subject Headings |

**Fig. 1.** NLLSH linked data page about the concept "Ontologies" [11].

Now that NLLSH has been published as linked data, it can be interlinked with the existing library linked open datasets. We plan to make use of the fact that NLLSH has links to LCSH and that LCSH is already linked to other library datasets − it has outgoing links to a number of linked data sources including Wikidata [1], FAST Linked Data[12] and the General Finnish Ontology YSO [3].

In the future, the published NLL datasets could be applied to interconnecting Latvia's cultural heritage information by describing museum and archive items with URIs from these datasets. Thus, the NLL ontology service could provide new possibilities for in-teroperability and data reuse for other Latvia's memory institutions and for memory institutions worldwide.

---

[10]  https://github.com/NatLibFi/Skosmos/wiki/ServingLinkedData
[11]  https://dati.lnb.lv/onto/nllsh/en/page/LNC10-000123185
[12]  http://experimental.worldcat.org/fast/

# 3 Conclusion

NLLSH authority data that forms the basis of the National Thesaurus of Latvia has been transformed into SKOS and published as linked data. The NLL linked authority data consists of 3 datasets: topical terms, geographic names, and form/genre terms. These linked authority datasets can be used for interlinking cultural heritage information, for example, by referring to their concepts not just from library information systems but also from other memory institutions such as archive and museum data. While preparing datasets for publishing as linked data, NLLSH data was enriched with linked record identifiers and language codes. We plan to provide regular updates to NLLSH linked data and to publish a related dataset – Latvia's National Bibliography – as linked data. Future work includes exploring options for automated addition of language codes, adding links to various cultural heritage information systems, and linking to other external linked data sets.

**Acknowledgments.** This work was supported by the Latvian Council of Science project no. lzp-2019/1-0365 "Latvian Memory Institution Data in the Digital Space: Connecting Cultural Heritage".

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
