# OpenReview forum: "National Library of Latvia Subject Headings as Linked Open Data"
_eswc-conferences.org/ESWC/2021/Conference/Poster_and_Demo_Track — ESWC2021 P&D_

### Official Review · AnonReviewer3 · 2021-04-06
**Nice to know, but a bit slim on new knowledge**

**Rating:** 6
**Confidence:** 3

**Review:**

In this poster/demo, the authors explain that they have converted and are publishing the subject headings (which used to be only available as MARC 21 data) as linked open data.

While this is interesting to know. This isn't really a very new thing to do. This is also visible from the various tools they reused from other libraries. Only a smal adaptation was done to tailor it to their needs. Besides, it appears that only very little prior work is included as references in the paper. There have been many efforts to make such data available, but they are not contrasted with the presented approach.

One aspect I suggest to include in the work is more of an explanation of what people can do with the data, perhaps with a concrete example. Currently, there is only a vague description of things that might well be possible (e.g., interlinking cultural heritage information).

**Anonymity:**

Yes, I would like my review to remain anonymous.

---

### Official Review · AnonReviewer1 · 2021-04-14
**NLL Linked Dataset: Good fit for ESWC, but discussions on the possible usage of the dataset and a rationale on the concept mappings are needed.**

**Rating:** 6
**Confidence:** 4

**Review:**

Review:
The authors present the National Library of Latvia linked dataset. The National Library of Latvia has published its subject headings in three datasets, topical headings, geographic names and form/genre terms as Marc21 data. As part of their work, the authors added unique record identifiers and language labels. The authors converted the data to SKOS concepts and published all three datasets using Skosmos.

The paper fits the conference topic.  The contribution of the work is sufficient for a poster paper and can be useful for the community. However, numerous authority controlled data already exist and it would have been useful to have a brief discussion on the specific contribution of the Latvian dataset to the community. What are concepts that are specific to Latvia which are not yet existing in other authority files? How can the created datasets now be used? The description of the conversion process itself is rather long and could be shortened since it is straightforward. In the example, the authors present the concept “Acteki”, which already exists in other authority files. To which extent does this concept differ from the concept “Aztecs” at the Library of Congress, since it is only considered a skos:closeMatch? Why are all external links a close match only? What are the criteria for these mappings and how do the authors evaluate the mapping quality? More exact mappings would also increase the usefulness of the dataset for the community. Furthermore, there is a discrepancy between the concept displayed in Figure 1 and the concept in the actual dataset. Please adapt the figure to match the data.

Strong points:
Good contribution
The dataset can be useful for the community

Weak points:
No discussion on mapping decisions and mapping quality
Weak discussion of the impact and usage of the dataset for the community

Overall, I think the poster paper is a good fit for ESWC, but in my opinion a more elaborate discussion on the possible usage of the dataset and a rationale on the concept mappings are needed.


**Anonymity:**

Yes, I would like my review to remain anonymous.

---

### Official Review · AnonReviewer2 · 2021-04-16
**Interesting work of offering librarian info as linked open data.**

**Rating:** 5
**Confidence:** 4

**Review:**

The paper National Library of Latvia Subject Headings as Linked Open Data explains how the authors constructed NLLSH Linked Data by converting MARC 21 authority records into NLL authority linked data.

PROS

The paper is definitely of relevance to the Semantic Web community since it presents a good example of transforming librarian info as linked open data. The presented approach is clear and relatively well written.

CONS

The presented 3 steps path from MARC 21 authority records to NLL authority linked data is rather straightforward. No lessons learned or interesting insights from accomplishing these 3 steps are given. What should be the main message for the reader?

SUMMARY

I am not against posters/demos that are not advanced in terms of new methods, etc. My main concern is the paper lacks insides from the practical exercise that the authors did to offer librarian info as linked open data. Thus, I have to suggest to reject the paper.


**Anonymity:**

Yes, I would like my review to remain anonymous.

---

### Decision · Program_Chairs · 2021-04-19

Accept